# Formation and Evaluation of a Two-Phase Polymer System in Human Plasma as a Method for Extracellular Nanovesicle Isolation

**DOI:** 10.3390/polym13030458

**Published:** 2021-01-31

**Authors:** Maria Slyusarenko, Nadezhda Nikiforova, Elena Sidina, Inga Nazarova, Vladimir Egorov, Yuri Garmay, Anastasiia Merdalimova, Natalia Yevlampieva, Dmitry Gorin, Anastasia Malek

**Affiliations:** 1Subcellular Technology Lab, N.N. Petrov National Medical Research Center of Oncology, 197758 St. Petersburg, Russia; slusarenko_masha@mail.ru (M.S.); niki2naden_ka@mail.ru (N.N.); sidina@mail.ru (E.S.); oblaka12@mail.ru (I.N.); 2The Faculty of Physics, Saint-Petersburg State University, 199034 St. Petersburg, Russia; yevlam_2007@mail.ru; 3Oncosystem Ltd., 121205 Moscow, Russia; 4Department of Molecular and Radiation Biophysics, Petersburg Nuclear Physics Institute Named by B. P. Konstantinov of National Research Center “Kurchatov Institute”, 188300 Gatchina, Russia; egorov_vv@pnpi.nrcki.ru (V.E.); yuri.from.spb@gmail.com (Y.G.); 5Center for Photonics and Quantum Materials, Skolkovo Institute of Science and Technology, 121205 Moscow, Russia; Anastasiia.Merdalimova@skoltech.ru (A.M.); D.Gorin@skoltech.ru (D.G.)

**Keywords:** polyethylene glycol, dextran, extracellular nanovesicles, two-phase polymer system, plasma, diagnostic

## Abstract

The aim of the study was to explore the polyethylene glycol–dextran two-phase polymer system formed in human plasma to isolate the exosome-enriched fraction of plasma extracellular nanovesicles (ENVs). Systematic analysis was performed to determine the optimal combination of the polymer mixture parameters (molecular mass and concentration) that resulted in phase separation. The separated phases were analyzed by nanoparticle tracking analysis and Raman spectroscopy. The isolated vesicles were characterized by atomic force microscopy and dot blotting. In conclusion, the protein and microRNA contents of the isolated ENVs were assayed by flow cytometry and by reverse transcription followed by quantitative polymerase chain reaction (RT-qPCR), respectively. The presented results revealed the applicability of a new method for plasma ENV isolation and further analysis with a diagnostic purpose.

## 1. Introduction

Circulating plasma has a complex composition including electrolytes, cyclic compounds, nucleic acids, proteins, lipoproteins of various densities, glycosides, and other compounds. Extracellular nanovesicles (ENVs) are minor components of plasma. Their quantitative and qualitative patterns are still poorly understood, but the emerging biological significance of ENVs is attracting growing scientific interest. It is now assumed that the population of circulating vesicles mediates the interaction of anatomically distant cells [1], and this population is extremely heterogeneous [2]. As evaluated by the original technology of cryo-electron microscopy of vesicles labeled with gold particles, plasma contains membrane formations of various morphologies, including spherical vesicles with a diameter of 30–1000 nm, tubular vesicles with a length of 1–5 μm, and membrane fragments up to 6–8 µm, with the former accounting for >95%, the latter accounting for <5%, and the third accounting for <0.5% of the entire population [3]. Moreover, 1 μL of plasma includes approximately 5 × 10^5^ spherical vesicles and 2 × 10^3^ tubular vesicles [3]. Other studies have presented quite different results of plasma ENV quantitation. For example, the concentration of plasma nanovesicles measured by nanoparticle tracking analysis (NTA) varied from 10^11^ to 10^12^ particles per mL [4]. Nanoflow cytometry (nFCM) allowed the detection of 10^7^ to 10^13^ exosome-like vesicles per 1 mL of plasma, depending on the method applied for isolation [5]. When ENVs were isolated by the standard ultracentrifugation procedure, the concentration of plasma ENVs differed by 2–3 orders of magnitude [4,5]. Such obvious discrepancies in the quantification of plasma ENVs may reflect the imperfection of both the isolation methods and the calculation technologies.

The isolation of nanovesicles from plasma is indeed a nontrivial task that has led to the development of many methods [6,7]. The current approaches can be generally divided into two main groups: methods based on specific physical properties of ENVs and methods based on the chemical characteristics of the vesicular membrane. The first group of methods includes ultracentrifugation as a “gold standard” [8], with various modifications (speed, time, gradient density medium) and ultrafiltration. More advanced approaches are based on microfluidic effects and nanoparticle flow fractionation phenomena [9]. The second group of methods is based on more or less specific binding of the components of the vesicular membrane with various ligands, including protamines [10], aptamers [11], antibodies [12], etc. This interaction leads either to the “fixation” of vesicles to a solid matrix or to the formation of multi-vesicular complexes, which are easily precipitated by low-speed centrifugation. The first group of methods makes it possible to obtain fairly pure ENV preparations from plasma, but these methods are relatively laborious and associated with losses of vesicles during the isolation process. The methods of the second group have one common drawback—there is no known vesicular marker that would be presented exclusively on ENVs and on all ENVs. Therefore, ENVs isolated by any of the affinity methods represent a certain fraction, which cannot accurately reflect the qualitative and quantitative composition of the entire population of plasma ENVs. Thus, the existing technologies for plasma ENV isolation have a number of limitations that compromise the results of further analysis and justify efforts to develop new approaches.

The phenomena of aqueous two-phase systems (ATPSs) formed by mixing two polymers can provide a promising approach for the isolation of ENVs from biological fluids. For the first time, the separation of polymer mixtures into two phases was shown for natural polymers, agar and gelatin [13]. Later, this phenomenon was applied to address issues of analytical chemistry [14]. Meanwhile, the nature of the ATPS phenomenon was studied more deeply. The van der Waals forces and electrostatic and hydrophobic mechanisms of intermolecular interactions are thought to be involved in ATPS formation [15]. The first attempt to apply ATPS for isolation of ENVs from biological fluids was done by Shin and co-authors [16] in 2015. In this study, ATPS was composed by dextran (DEX) and polyethylene glycol (PEG) in artificial quantity-defined ENV-polymer mixture that allowed them to analyze in detail the distribution of vesicles and proteins. Later on, this group explored DEX–PEG ATPS for the isolation of ENVs from urine [17,18]. The authors of these studies isolated ENVs with 100% efficiency, claiming excellent efficacy of the method. Recently, another group successfully explored the efficacy of DEX–PEG ATPS for the isolation of ENVs from different biological sources including plasma [19]. However, authors of this report combined plasma with premixed polymer solution to achieve phase separation in the resulting mixture.

There is a reason to believe that the phase separation of polymers can also be achieved directly in plasma, and it may present a cost-effective and user-friendly method for plasma ENV isolation. Plasma ENVs sized approximately 100 nm are supposed to be enriched by exosomes that attract particular interest as potential multimolecular markers for clinical diagnostics. Therefore, the aim of the present study was to explore the two-phase polymer (DEX–PEG) system formation directly in human plasma and to evaluate efficacy of ENV isolation by such plasma two-polymer system (PTPS).

## 2. Materials and Methods

### 2.1. Biological Samples

The study was approved by the local ethics committee of N.N. Petrov NMRC of Oncology (10.11.2017, protocol AAAA-A18-118012390156-5) and conducted in accordance with the ethical guidelines outlined in the Declaration of Helsinki. Plasma was obtained from healthy donors in the blood transfusion department. All participants signed a voluntary and informed study participation form.

### 2.2. Components of PTPS

DEX 20 kDa (Serva, Heidelberg, Germany), DEX 200 kDa (Serva, Heidelberg, Germany), DEX 450–650 kDa (Sigma-Aldrich, St. Louis, MO, USA), PEG 6 kDa (Serva, Heidelberg, Germany), PEG 15 kDa (Loba Chemi, Mumbai, India), PEG 20 kDa (Loba Chemi, Mumbai, India), PEG 35 kDa (Sigma-Aldrich, St. Louis, MO, USA).

### 2.3. Plasma Preparation

Blood was collected in a BD Vacutainer spray-coated ethylenediaminetetraacetic acid (EDTA) tubes, and plasma was immediately separated from the blood, aliquoted, and stored at −80 °C. Before use, plasma was slowly thawed at +4 °C. In order to remove cells, cellular detritus and large protein complex plasma was centrifuged: 300× *g*—10 min, 2000× *g*—20 min, 10,000× *g*—40 min.

### 2.4. ENV Isolation by Differential Centrifugation (UC)

The plasma (1.5 mL) cleared from cellular debris and large protein complexes was mixed with phosphate buffered saline (PBS) in a ratio 1:1. ENVs were sedimented by ultracentrifugation according to the classical procedure [8], with minor modifications using an Optima XPN 80 ultracentrifuge (rotor 70.1 Ti/k-factor 36). Briefly, the 50% plasma solution (3 mL) was centrifuged at 110,000× *g* for 2 h, the supernatant was removed, and the precipitate was dissolved in PBS (3 mL). The solution was centrifuged again at 110,000× *g* for 2 h. The supernatant was removed, and the pellet was dissolved in 500 μL of PBS.

### 2.5. ENV Isolation by Plasma Two-Phase Polymer System (PTPS)

The polymers, PEG and DEX, were dissolved in plasma (1.5 mL) at the desired concentration. In parallel, the same quantities of polymers were dissolved in PBS (1.5 mL) to prepare aqueous mixture herein referred to as protein-depleting solution (PDS). Two tubes containing plasma and PDS were centrifuged at 1000× *g* for 10 min to speed up partition of polymer solutions into lower phase (LP1) and upper phase (UP1). To deplete LP1 with plasma protein, the upper phase (UP1) was replaced by PDS, and solutions were mixed and re-separated into LP2 and UP2 by centrifugation at 1000× *g* for 10 min. To see the effect of the third re-separation, this procedure was repeated twice. To analyze the composition of two phases, the upper phase was taken as it is, while the lower phase was dissolved in PBS up to 500 μL.

### 2.6. ENV Isolation by Commercial Kits

ENVs were isolated from plasma with two commercial kits EXO-Prep (HansaBioMed, Tallinn, Estonia) and Total Exosome Isolation Kit (Invitrogen, Waltham, MA, USA). In each case, isolation was performed from 1.5 mL of plasma according to the manufacturer’s protocols.

### 2.7. Viscosimetry and Dynamic Light Scattering (DLS)

To evaluate particle size distribution in LPs of PTPS, the volumes of lower phases have been increased with PBS up to 1 mL. The dynamic viscosity values were obtained for all tested samples using an automatic microviscometer Lovis 2000 M/ME (Anton Paar GmbH, Ostfildern, Germany) at 298 K. Four measurements followed by averaging were performed for each sample. Dynamic light scattering was performed with a Nanotrac Wave II instrument (Microtrac Inc., Montgomeryville, PA, USA) at 25 °C; the signal accumulation time was 30 s. The results were processed in FLEX Software (Microtrac Inc., Montgomeryville, PA, USA) taking into account the earlier determined values of the suspension viscosity. Each sample was assayed in triplicate, and results were averaged.

### 2.8. Nanoparticle Tracking Analysis (NTA)

The measurements were carried out with a Nanosight NS300 analyzer (Malvern Panalytical, Malvern, UK). The spectra were processed using Nanosight NTA 3.2 Software. Camera level: 14, shutter slider: 1259, slider gain: 366, threshold level for LP—5, for UP—6. Each sample was pumped through the analyzer observation chamber to make 4–5 measurements at different microvolumes. Each measurement lasted 60 s, which corresponded to 1498 frames.

### 2.9. Atomic Force Microscopy (AFM)

The measurements were carried out using a scanning probe microscope NT-MDT Solver Bio (NT-MDT, Moscow, Russia) in a semi-contact mode, probe NSG01_DLC (NT-MDT, Moscow, Russia). Samples were applied to the surface of mica (SPI Supplies, West Chester, PA, USA), immediately after removing the top layer, incubated for 30 s, washed twice with distilled water, then dried using compressed air. Image processing was performed using the Gwyddion 2.54 (gwyddion.net) and ImageAnalysis (NT-MDT, Moscow, Russia) programs.

### 2.10. Raman Spectroscopy (RS)

Before measurements, samples (5 μL) were applied to quartz glass and dried at room temperature. A Raman spectrometer was used for measurements: LabRAM HR Evolution (HORIBA France SAS, Longjumeau, France), diffraction grating: 600 lines/mm, objective: Olympus MPlan, 50×, laser wavelength: 633 nm, exposure time: 50 s or 120 s depending on the severity (intensity) of the peaks of Raman scattering. The obtained Raman spectra were processed in MATLAB: the low-frequency component was removed using the BEADS algorithm [20], the spectra were smoothed using the Savitsky–Golay filter [21], and normalized to the intensity of the maximum peak to the interval [0; 1]. Before examining samples of different phases of the PTPS, the minimum detectable concentrations of polymers were determined: for DEX: 1.5 × 10 ^−3^ g/dL, for PEG: 7 × 10 ^−3^ g/dL.

### 2.11. Dot Blotting

Samples (2 μL) were transferred onto a polyvinyl membrane, blocked for 45 min in 5% bovine serum albumin (BSA) solution in Tris-saline buffer (20 mm Tris-HCl, 150 mm NaCl, pH 7.5), then incubated in BSA solution (0.1%) and tween-20 (0.05%) in Tris-saline buffer with primary antibodies to the CD63 (ab68418, Abcam, Cambridge, MA, USA), CD9 (ab18241, Abcam, Cambridge, MA, USA), HSP70 (ab181606, Abcam, Cambridge, MA, USA), CANX (ab238078, Abcam, Cambridge, MA, USA), and HAS (4T24, HyTest, Turku, Finland) at a dilution of 1:5000 at +4 °C overnight. Blots were visualized using secondary antibodies conjugated with peroxidase (ab6721 and ab6789, Abcam, Cambridge, MA, USA), a Pierce ™ ECL Western Blotting Substrate kit (Thermo Fischer Scientific, Waltham, MA, USA) on an IBright FL1000 apparatus (Thermo Fischer Scientific, Waltham, MA, USA).

### 2.12. Isolation of RNA and RT-PCR Analysis of MicroRNA

The total RNA from vesicles (obtained by different methods from 1.5 mL of plasma) was isolated using the Lira+ kit (Biolabmix Ltd., Novosibirsk, Russia) intended for the isolation of small RNA from cells and cell suspensions according to the protocol of the manufacturer. Elution of RNA was carried out in all cases in a volume of 30 μL. Quantitative analysis of two microRNAs was performed by two-tailed RT-qPCR technology using kits AL451a-5p and AL126-3p (Algimed Techno LLC, Minsk, Belarus) according to the manufacturer’s protocol. The CFX96 Real-Time PCR amplifier (Bio-Rad, Hercules, CA, USA) was used. Each reaction was carried in triplicate, and the values of the threshold cycles (Ct) were averaged.

### 2.13. Flow Cytometry (FC)

The Exo-FACS kit (HansaBioMed, Tallinn, Estonia) was used to attach ENVs to latex beads according to the manufacturer’s protocol. The exosomal marker CD63 was detected using an FITC-conjugated antibody (ab18235, Abcam, Cambridge, MA, USA). Flow cytometry data were obtained on a CytoFLEX analyzer (Beckman Coulter, Indianapolis, IN, USA) equipped with an argon laser with a wavelength of 488 nm for measuring forward light scattering (FLS) and side scattering (SS) and detecting the FITC signal.

## 3. Results

### 3.1. Optimization of the PTPS Parameters: Polymer Molecular Mass and Concentration

As reported previously [17,19], aqueous solutions of 25–45 kDa PEG and 450–650 kDa DEX form a two-phase system that can be used to isolate ENVs from biological fluids. We reproduced the reported system using human plasma as the solvent; however, the formation of two phases was not observed. Apparently, the complex composition of plasma interfered with the polymer solution and affected the conditions required for phase separation.

Therefore, we decided to systematically explore various molecular masses (MMs) and concentrations of polymers. First, we tested twelve systems by combining PEG (6, 15, 20, and 35 kDa) and DEX (20, 200, and 450–650 kDa). The concentrations of polymers in each pair were defined within a range from 1% to 7% (*w*/*v*) on the basis of phase diagrams described for these or similar systems formed in aqueous solutions [13,14,22,23]. In most cases, we observed partitions of polymers in plasma-based solutions; however, lower phases were either hardly dissolved or formed polydisperse suspensions. Bright lower phases (LP) easily dissolved in PBS were formed in PTPS with PEG (20 kDa) and DEX (450–650 kDa). The DLS analysis revealed a unimodal size distribution of particles in LPs; however, the average size and distribution range varied considerably depending on the polymer concentrations.

Next, we kept selected MMs of polymers and tested systems of equal concentrations from 1% to 5%, as well as unequal concentrations of DEX (from 1% to 4.5%) and PEG (from 1.5% to 5%). The reproducible enrichment of LP with a unimodal fraction of plasma components with sizes corresponding to the size of exosomes (~100 nm) was obtained using PTPS containing PEG (20 kDa) 3.5% and DEX (450–650 kDa) 1.5%, as shown in Figure 1.

The use of other polymer concentrations, for instance, PEG 3%–DEX 3% and PEG 2%–DEX 2%, led to the isolation of larger plasma components or to the aggregation of small particles. Based on the obtained results, the system formed by PEG (20 kDa) 3.5% and DEX (450–650 kDa) 1.5% was considered optimal and was used in further experiments.

As reported previously [17,19], the single partitioning of PEG/DEX ATPS is not enough to completely separate vesicles and proteins of biological fluids. For instance, to reduce the amount of plasma proteins in LP1, it was suggested to replace UP1 with fresh PEG solution or PEG/DEX mixture and repeat phase separation [19]. With the goal of isolating plasma ENVs, we attempted to reproduce this procedure with a plasma two-phase polymer system (PTPS) and to further investigate the properties of the resulting phases. In general, the two-step procedure is schematically shown in Figure 2.

The clarification of the upper phase was observed in the process of two-stage separation. The tubes with PTPS formed by PEG (20 kDa) 3.5% and DEX (450–650 kDa) 1.5% in Steps 2 and 6 of the procedure are imaged in Figure 3.

### 3.2. Plasma Two-Phase Polymer System Characteristics

To gain insight into the composition of the two-polymer system, we used nanoparticle tracking analysis (NTA) and Raman spectroscopy (RS). The analysis of UP1 and LP1 obtained after the first separation is shown in Figure 4A. The figure also includes the results of the analysis of plasma ENVs isolated by the standard UC technique as a reference. UP1 was measured as it is, while LP1 and UC-isolated ENVs (UC-ENVs) were resuspended in PBS up to 500 μL before measurement. The LP1 and UC-ENV had similar particle size distribution profiles, with major peaks corresponding to 102 and 105 nm. Even the minor “peaks” at approximately 200 and 250 nm observed in the UC-ENV samples were reflected by somewhat more representative “peaks” in LP1. UP1 contained only barely detectable amounts of small particles (30–50 nm), which might account for protein complexes or serum lipoproteins. Since the sizes of the major plasma protein albumin (5–6 nm) are much lower than the NTA detection threshold, the amount of plasma proteins in UP1 cannot be estimated by this method.

Next, we evaluated the composition of UP2 and LP2, as shown in Figure 4B. The profile of the nanoparticle size distribution in LP2 was not changed considerably compared to LP1, and the major “peak” in LP2 was observed at 98 nm. Surprisingly, re-separation resulted in considerable alteration of the upper phase content. A heterogeneous mix of particles from 50 to 270 nm was detected in UP2 that could reflect “transition” of lipoproteins, vesicles, or their aggregate from LP2 to UP2 during repeated phase partitioning. To estimate losses of exosomal size ENVs during re-separation, the number of particles within the dispersion from 50 to 150 nm were calculated by integration in LP1 and LP2 (range of integration is shown by dotted line in Figure 4)**.** Thus, LP1 contained 3.2 × 10^11^ particles/mL while LP2 contained 3.0 × 10^11^ particles/mL, that revealed non-significant losses of ENVs due to re-separation.

Raman spectroscopy was applied to explore the chemical compositions of the phases by evaluation of their characteristic spectra. Figure 5 shows the Raman spectra of “pure” PTPS components: DEX, PEG, and UC-isolated ENVs dissolved in PBS and four phases of PTPS (UP1, LP1, UP2, and LP2). Characteristic peaks of “pure” PTPS components and their likely assignments reported previously [24,25,26,27,28,29,30,31,32,33,34,35] are listed in Appendix A. For example, UC-ENV-specific peak at 1004 cm^−1^ may reflect the presence of an essential α-amino acid phenylalanine [24], however enrichment of this compound in plasma ENV compared to total plasma was not detected [36]. The composition of tested solution included various plasma components and was too complex for precise assigning. Without identifying specific peaks, it is important to note that several characteristic peaks can be observed in spectra of UC-ENV and they are repeated in spectra of lower phases after first (LP1) and second (LP2) separation (red frame). This observation can support the phenomena of the concentration of the plasma vesicles into the dextran-formed lower phase of PTPS.

Raman spectra of pure DEX and PEG had also several characteristic peaks. As expected, the profiles of both the LP1 and LP2 phases had similarities to the DEX profile (marked by blue arrows), and these results were well reproducible. However, multiple analyses of the upper phases revealed different results, resembling either DEX or PEG profiles (marked by blue and green arrows). It became especially visible after the second phase separation. Two types of spectra, PEG-like with characteristic peaks shown by green arrows and DEX-like with characteristic peaks shown by blue arrows, were detected in UP2. To determine the nature of this observation, samples dried on the surface of quartz glass were visualized before Raman spectroscopy. The dried UP2 phase revealed a bubble-like texture, as shown in Figure 6. Raman spectra taken inside of these “bubbles” showed a DEX-like profile, while spectra measured outside of “bubbles” were similar to PEG spectra. This result revealed the presence of both polymers in upper phases and their tendency to segregate within them. Incomplete partitioning of the plasma two-phase polymer system during the applied procedure might influence the efficacy of plasma component separation.

Taken together, the results of NTA and RS confirm the phenomena of phase separation in plasma solutions of PEG (20 kDa) 3.5% and DEX (450–650 kDa) 1.5%. Particles of exosomal size are concentrated in the lower phase, and their Raman spectra are similar to the spectra of UC-isolated plasma ENVs. Repeated phase separation did not considerably change the nanoparticle content (NTA) or chemical composition (RS) of the lower phase; however, it resulted in the appearance of nanoparticles of different sizes in the upper phase. It is interesting to note that the second round of phase separation resulted in an appearance of differently-sized particles in the upper phase (NTA, Figure 4B: UP2) and DEX-characteristic peaks 546 and 922 cm^−1^ (RS, Figure 5: UP2) in its Raman spectra, and that both were not detected in UP1. However, the currently available data are not sufficient to explain this coincidence.

From a practical point of view, distribution of plasma proteins between phases of the two-phase polymer system is an important issue. Thus, total protein concentrations of two phases during three rounds of separation was assayed, as shown in Figure 7A. Taking in account that the upper phase after each round of separation accounted for about 90–95% of total volume of suspension, as shown in Figure 3, it can be considered that a substantial amount of plasma proteins had accumulated in UP and were removed from the system during consequent rounds of phase separation. Figure 7B demonstrates relative concentrations of major plasma proteins (albumin) in corresponding phases. It can be clearly seen that concentration of albumin had been stepwise reduced in the whole system and especially in LP; that may indicate clearance of ENV-containing PL from plasma proteins.

### 3.3. Characteristics of Isolated Vesicles

To confirm the exosomal nature of nanoparticles concentrated in the lower phase of PTPS, we evaluated their topology by atomic force microscopy (AFM) and analyzed a set of exosomal markers by dot blotting.

LP2 and UC-ENV as a control were scanned, as shown in Figure 8A, D. The particles concentrated in LP2 of PTPS are presented at different scan scales in Figure 8B,C. The image shows particles of two sizes: large particles (70–90 nm) probably corresponded to exosomes, and small particles (40–50 nm) could be dextran globules. Figure 8D shows a histogram of the particle distribution, which confirms the presence of two fractions of nanoparticles of different sizes. Moreover, the sizes of UC-ENVs coincide with the sizes of large particles, which argues in favor of enrichment of LP2 with exosomes.

Next, we compared the presence of the exosomal markers in different phases of PTPS. However, standard markers (CD63, CD9, and HSP70) were not detectable in neither UP1 nor UP2 by conventional approaches. The results of the LP1, LP2, and UC-ENV analysis by dot blot are presented in Figure 9. All samples were obtained from 1.5 mL of plasma and diluted up to 500 µL. The negative marker CANX was not detected in any samples. All three exosomal markers (CD9, CD63, and HSP70) were presented in LP1. Re-separation of the phases of PTPS resulted in an obvious reduction of CD63-positive nanoparticles, while the amounts of CD9 and HSP70 were only slightly decreased. This demonstrates either unequal representation of classical exosomal markers on plasma ENVs or selective leakage of CD63-positive ENVs from the lower phase during re-separation. However, the ratio of exosomal marker concentrations in LP2 was rather similar to that in UC-isolated ENVs that again argued in favor of concentration of plasma ENVs in LP2.

According to the AFM and dot blotting results, nanoparticles concentrated in lower phases of PTPS had morphology and protein marker profiles similar to those of ENVs isolated from plasma by standard ultracentrifugation. Re-separation of PTPS results in a reduction in CD63 content in the lower phase.

### 3.4. Analysis of Protein Surface Markers and miRNAs in Isolated Vesicles

A practically important issue is the possibility of using plasma ENVs isolated by PTPS for further analysis. To address this question, we assayed vesicular surface markers by flow cytometry and vesicular miRNA RT-qPCR.

We used latex particles (beads) with a size of 4 µm. ENVs were nonspecifically attached on the bead surface according to the manufacturer’s protocol. Detection of exosomes on the surface of the particles was carried out using fluorescently labeled antibodies to the exosomal marker CD63, as shown in Figure 10. To exclude false positive results due to auto-fluorescence of latex particles or PTPS components, corresponding controls were carried out: non-treated particles (A), particles after incubation with PEG (B), and particles after incubation with DEX (C). To exclude false positive results due to nonspecific interactions of fluorescently labeled antibodies with particles or PTPS components, the corresponding controls were analyzed without incubation of the particles with exosomes (D, E). A distinct shift in the intensity of the fluorescent signal was observed in the sample of particles after their incubation with ENVs isolated by PTPS (F) or by UC (G) as a control and following staining with anti-CD63-FITC antibodies.

Next, we evaluated the efficiency of quantitative analysis of microRNAs from ENVs isolated by PTPS. In parallel, two commercial kits based on chemical precipitation technology were used. Both EXO-Prep (HansaBioMed, Tallinn, Estonia) and the Total Exosome Isolation Kit (Thermo Fischer Scientific, Waltham, MA, USA) were used for the isolation of exosomes from plasma and subsequent RT-PCR analysis of exosomal RNA. Five samples of donor plasma were combined in one pool. ENVs were isolated from 1.5 mL of plasma by each method in triplicate. After total RNA isolation, RT-PCR analysis of two microRNAs (miR-451 and miR-126) in the three samples was carried out by reverse transcription with a two-tailed primer (two-tailed RT) and subsequent PCR according to the instructions of the ALLMIR Kit producer. Additionally, these kits allowed us to test in parallel a set of serial dilutions of synthetic analogs of target miRNA. The results are shown in Figure 11.

In both cases, miRNAs (miR-451 and miR-126) in biological samples were detected within the areas of linear ratios of synthetic miRNA concentrations and threshold cycles (Ct) that reflected sufficient analytic properties of systems. Both miR-451 and miR-126 were detected in samples of ENVs isolated by PTPS and by commercial kits at comparable or even higher efficacy.

Taken together, the results of both flow cytometry and RT-qPCR demonstrated that ENVs isolated by PTPS are suitable for further analysis in terms of protein surface markers and miRNA content.

## 4. Discussion

Our study is an attempt to apply the phenomena of a two-phase polymer system to the task of plasma ENV isolation. The main advantages of this technology are its low cost, technical simplicity, and, consequently, its scalability. The ENV isolation process took in whole less than 2 h and did not require specific equipment, expensive reagents, or laboratory skills. However, the implementation of the proposed technology in laboratory or even clinical practice still requires the addressing of two fundamental issues. First, the influence of human plasma components on the behavior of the two-polymer system and the principles of phase separation of various plasma compounds are still poorly understood. Second, a detailed analysis of the population of plasma ENVs isolated by a particular PTPS should be performed to evaluate how representative this fraction is in terms of clinical diagnosis.

Most of the tested two-polymer systems were investigated previously, and their behavior in aqueous solutions was described by phase diagrams [13,14,22,23,37]. Due to the complex composition of human plasma, the plasma two-phase polymer system behaved differently. In most the polymers mixtures tested, we observed phase separation; however, the distribution of plasma components between the two polymer-formed phases varied considerably. The difference was especially noticeable in the volume, color, and consistency of the lower phases. As established many years ago, the separation of proteins in the presence of PEG is strongly affected by the charge of the polymer and protein molecules [38]. Recently, the specific pattern of PEG molecule distribution around the surface of different plasma proteins was modeled using molecular dynamic stimulation [39]. These results may explain the variable efficacy of the interaction between plasma polymers and PEG molecules of different MMs and, consequently, the different patterns of plasma protein partitioning in different PTPSs. Since the behavior of the plasma two-polymer system is defined by two polymers, as well as multiple plasma proteins, this behavior is difficult to extrapolate from a similar aqueous two-polymer system, and more research is needed to clarify the corresponding phase diagram and principles of plasma protein phase separation. In our study, we compared different PTPSs composed of DEX and PEG and selected one (PEG (20 kDa) 3.5% and DEX (450–650 kDa) 1.5%) that allowed the concentration of nanoparticles approximately 100 nm in size in the lower DEX-formed phase. However, our results are not sufficient to explain why this system worked better than others and why change of the concentration ratio of these polymers resulted in the dramatic change of size of particles concentrated in LP.

Since previous studies indicated the possibility of reducing the amount of plasma proteins in the lower phase by repeated phase separation, we reproduced this procedure and assayed the phase composition. NTA revealed no significant reduction in the quantity of 100 nm particles in LP2 compared to PL1, which is in accordance with published data [19]. Interestingly, NTA of the upper phases revealed significant alteration of the nanoparticle size distribution profile as a result of repeated phase separation—UP1 contained a barely detectable number of particles 30–50 nm, while UP2 included many more particles of different sizes, as shown in Figure 4. Raman spectroscopy revealed the similarity of the spectra of UC-ENVs and both LP1 and LP2, which confirmed the presence of exosomes in lower phases. However, the heterogeneity of LP2 was observed by microscopy of dried samples and by Raman spectroscopy. Further research is needed in order to understand whether NTA-detected particles and the DEX-like area detected by RS in LP2 are just a coincidence, or if they are somehow associated. Importantly, the reduction of the total protein and the albumin concentration in the LP after repeated phase separation was detected by Bradford assay and dot blot. On the basis of the obtained results, it can be concluded that repeated phase separation resulted in clearance of the low phase from an excess of major plasma proteins.

The results obtained in our study (NTA and AFM) allowed us to claim only the relative physical homogeneity of isolated ENVs. The results of the dot blot assay revealed the presence of classical exosomal markers (tetraspanins CD63 and CD9 and members of the hot shock protein family HSP70) and the absence of the endoplasmic reticulum component calnexin (CANX). This result confirmed the enrichment of the isolated ENV population by exosomes but did not exclude the presence of other plasma components of similar size. Finally, we demonstrated that ENVs isolated by PTPS are suitable for further analysis of their surface proteins by flow cytometry and miRNA by RT-qPCR. Previously, Shin and co-authors [17] analyzed the membrane protein PSMA by ELISA and PCA3 mRNA by RT-PCR in vesicles isolated by an aqueous two-polymer (DEX/PEG) system. Thus, dissolving these polymers directly in plasma allows the isolation of exosome-enriched ENVs in the quality and quantity appropriate for further analysis with diagnostic purposes.

## Figures and Tables

**Figure 1 polymers-13-00458-f001:**
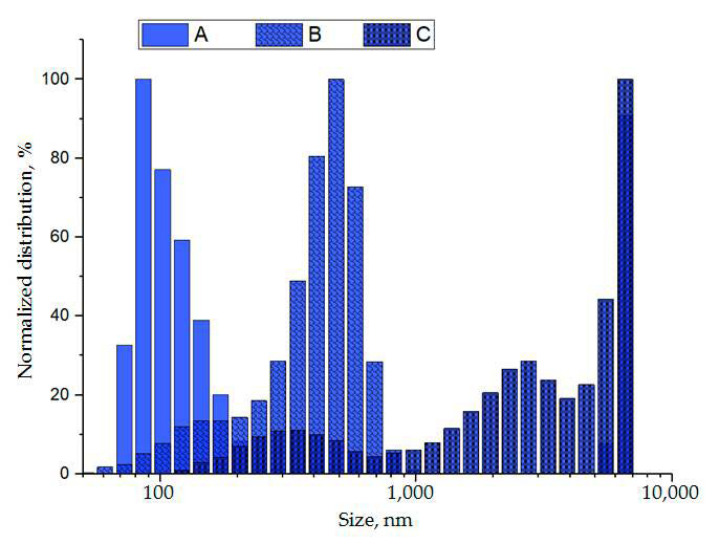
Size distribution of particles in lower phases (LP1) of plasma two-polymer systems (PTPS) formed by polyethylene glycol (PEG) (20 kDa) and dextran (DEX) (450–650 kDa) measured at different concentrations and dynamic viscosity values. A. 3.5% PEG–1.5% DEX, 1.0214 mPa·s; B. 3% PEG–3% DEX, 1.0516 mPa·s; C. 2% PEG–2% DEX, 1.0756 mPa·s.

**Figure 2 polymers-13-00458-f002:**
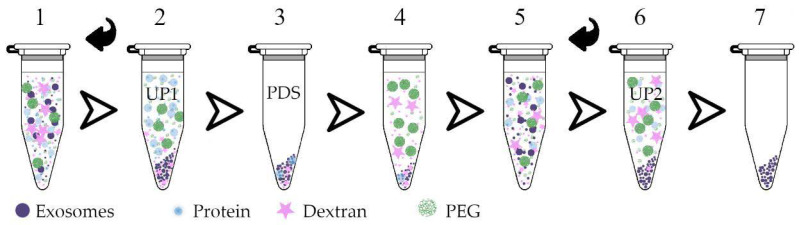
Steps of the PTPS-based extracellular nanovesicles (ENV) isolation protocol. (**1**) A mixture of plasma and polymers (dextran/polyethylene glycol); (**2**) two phases formed after mixing and centrifugation; (**3**) the upper phase “UP1”, containing PEG and proteins, is removed; (**4**) UP1 is replaced with a protein-depleting solution (PDS); (**5**) stirring; (**6**) the solution is again separated into two phases by centrifugation; (**7**) as a result, together with the formed upper phase “UP2”, protein residues are removed, while the lower phase contains dextran and ENVs.

**Figure 3 polymers-13-00458-f003:**
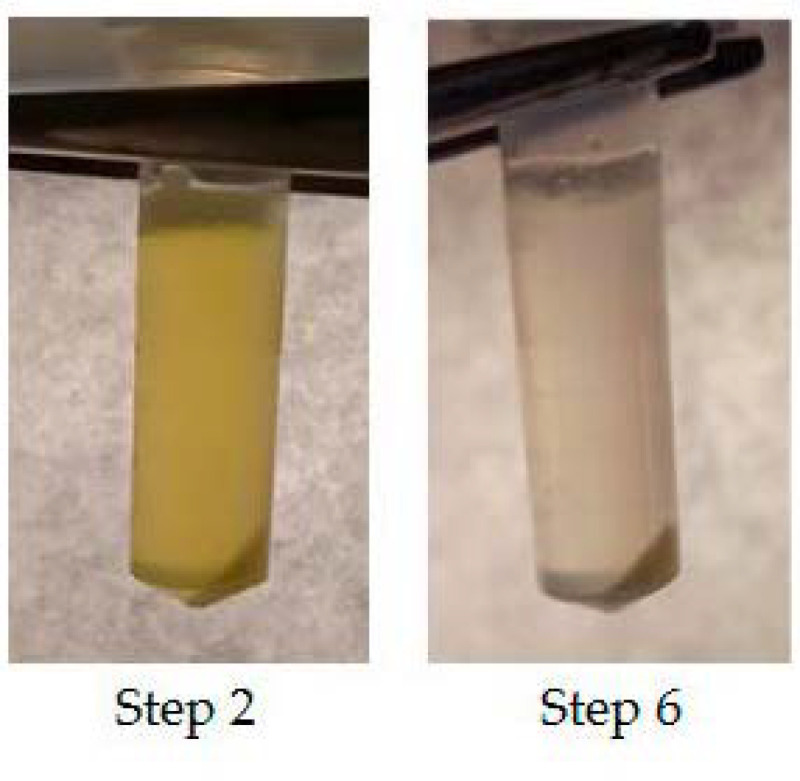
PTPS at Step 2 and Step 6 of the protocol.

**Figure 4 polymers-13-00458-f004:**
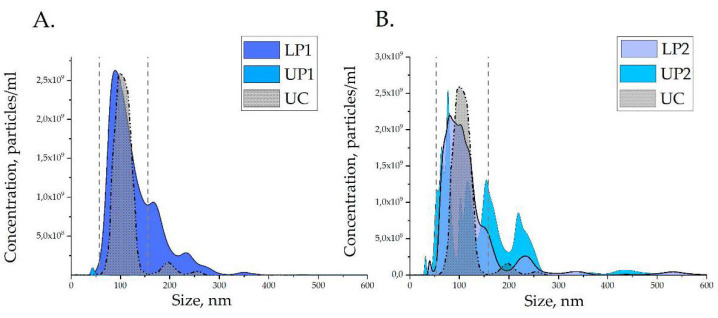
Nanoparticle size distribution in PTPS phases and extracellular nanovesicles isolated by ultra-centrifugation (UC-ENV) measured by nanoparticle tracking analysis NTA. (**A**) First phase separation; (**B**) second phase separation.

**Figure 5 polymers-13-00458-f005:**
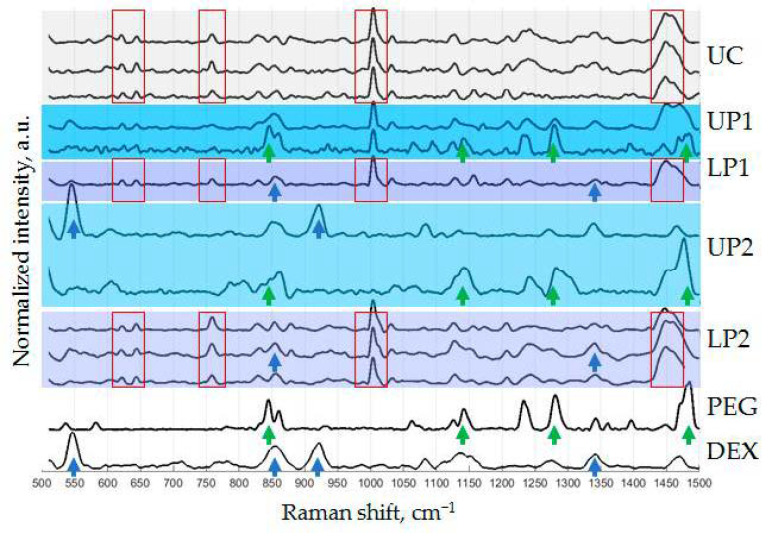
Raman spectra of PTPS phases and pure components. Measurements were repeated for samples that were supposed to be not homogeneous enough—these results are shown by multiple curves.

**Figure 6 polymers-13-00458-f006:**
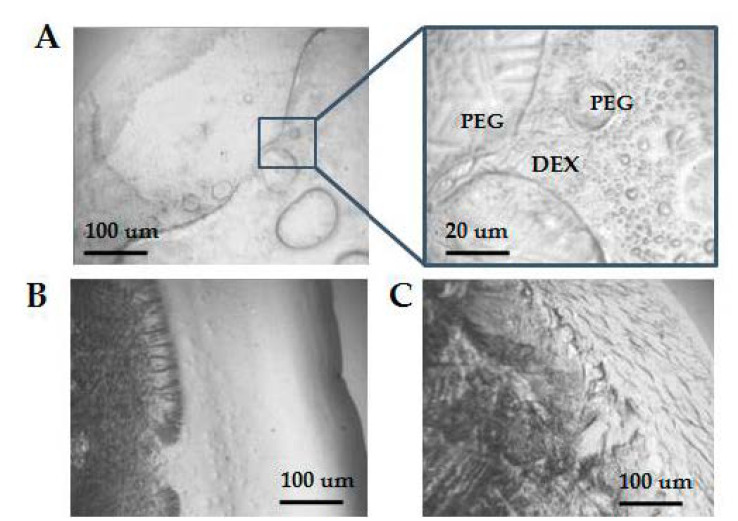
Area of dried drops of polymer mixtures and pure solutions visualized by light microscopy before Raman spectrometry. (**A**) Upper phase 2 UP2; (**B**) Dextran; (**C**) PEG.

**Figure 7 polymers-13-00458-f007:**
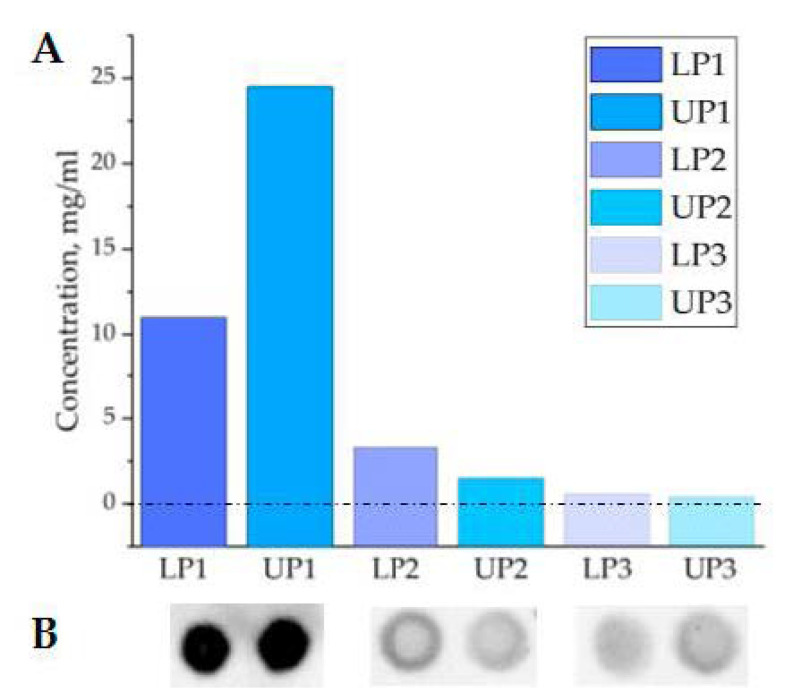
Protein concentration in two phases of PTPS during consequent separation. (**A**) Total protein concentration measured by Bradford assay. (**B**) Albumin quantity assayed by dot blotting.

**Figure 8 polymers-13-00458-f008:**
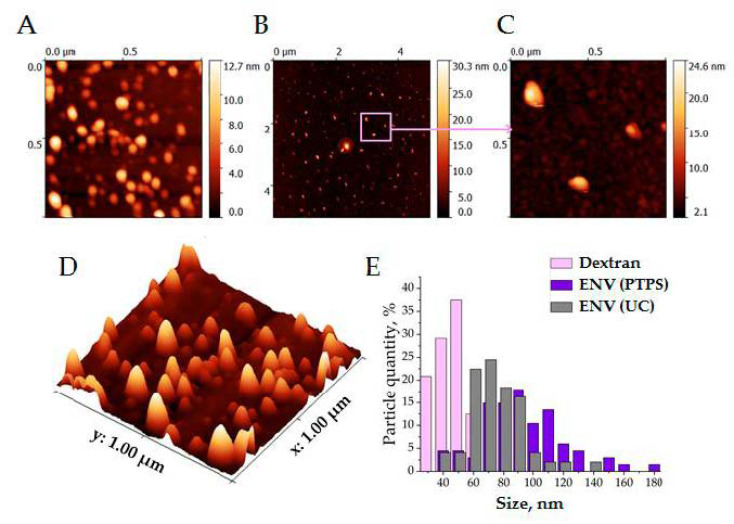
Particles concentrated in lower phase 2 (LP2) visualized by atomic force microscopy (AFM). (**A**,**D**) Images of ENVs obtained by ultracentrifugation as control; (**B**,**C**) images of ENVs isolated by PTPS; (**E**) the result of statistical processing of AFM images.

**Figure 9 polymers-13-00458-f009:**
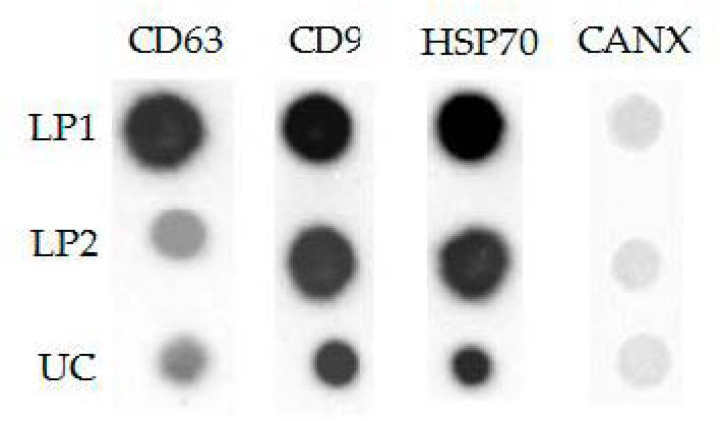
Exosomal marker evaluation in lower phases of PTPS and UC-ENV by dot blot.

**Figure 10 polymers-13-00458-f010:**
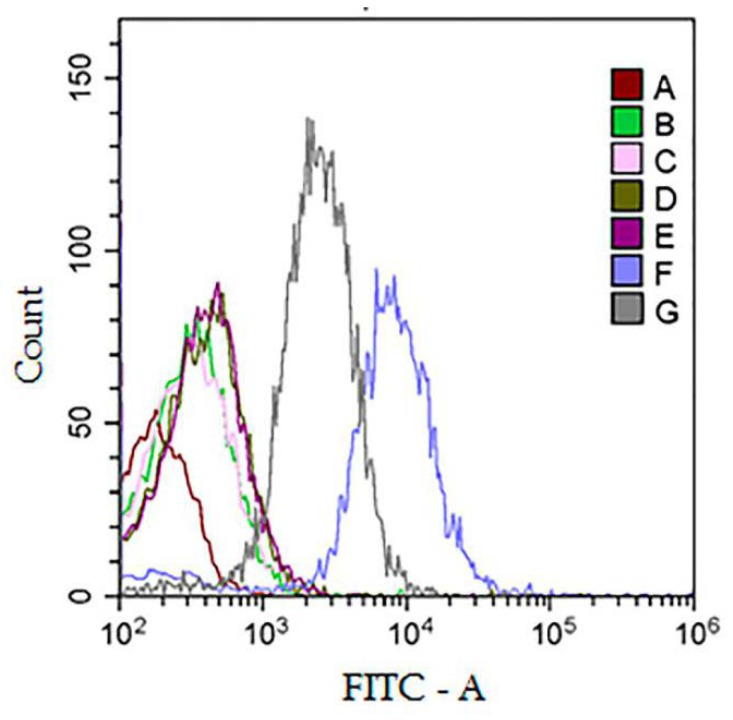
Flow cytometry analysis of ENVs isolated by PTPS. (**A**–**D**) Negative controls. (**A**) Latex particles (LPs); (**B**) LPs incubated with PEG; (**C**) LPs incubated with DEX; (**D**,**E**) LPs with PEG and DEX, respectively, incubated with an antibody against CD63-FITC; (**F**) LPs incubated with ENVs isolated by PTPS and labeled by an antibody against CD63-FITC; (**G**) LPs incubated with UC-ENV labeled by an antibody against CD63-FITC.

**Figure 11 polymers-13-00458-f011:**
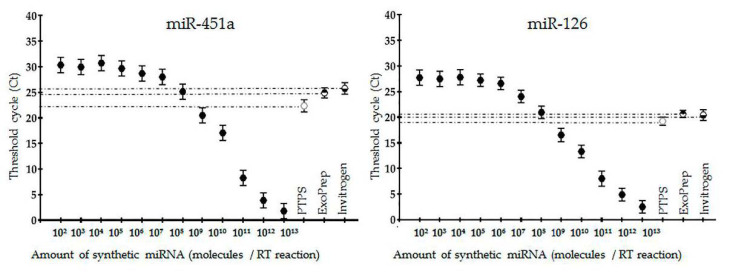
OT-qPCR analysis of miRNA in ENVs isolated by PTPS and commercial kits.

## Data Availability

Data available on request.

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
