# Peer review of "Formation and Evaluation of a Two-Phase Polymer System in Human Plasma as a Method for Extracellular Nanovesicle Isolation"

_polymers, 2021, doi:10.3390/polym13030458_

Round 1

Reviewer 1 Report

In ‘Formation and evaluation of a two-phase polymer system in human plasma as a method for extracellular nanovesicles isolation’, Slyusarenko et al. optimized a polymer mixture for phase separation of extracellular nanovesicles from human plasma. The nanovesicles were characterized for their material and biochemical properties.

Specific comments:

Lines 81-89: It would be helpful to the reader to provide some more detail about the previous work with the combination of the polymer mixture and plasma to clearly distinguish what is novel about the present study.

Lines 244-245: The term ‘picks’ is unclear.

Figure 5: Some further information about the different spectra should be provided in the caption. In particular, it is unclear what is being represented by the multiple curves, e.g., labeled UC.

Figure 6: Scale bars should be added instead of labels with the magnification.

Throughout, the abbreviation DXT is unclear.

Author Response

Authors are very grateful to the Reviewer for a careful analysis of our manuscript and important comments.

In ‘Formation and evaluation of a two-phase polymer system in human plasma as a method for extracellular nanovesicles isolation’, Slyusarenko et al. optimized a polymer mixture for phase separation of extracellular nanovesicles from human plasma. The nanovesicles were characterized for their material and biochemical properties.

Specific comments:

  1. Lines 81-89: It would be helpful to the reader to provide some more detail about the previous work with the combination of the polymer mixture and plasma to clearly distinguish what is novel about the present study.

The main novelty of our technique is dissolving of polymers directly in plasma and use of plasma as a solvent. We now extended the indicated paragraph to highlight this aspect.

  1. Lines 244-245: The term ‘picks’ is unclear.

Thank you. It’s corrected now.

  1. Figure 5: Some further information about the different spectra should be provided in the caption. In particular, it is unclear what is being represented by the multiple curves, e.g., labeled UC.

Thank you. It’s corrected now.

  1. Figure 6: Scale bars should be added instead of labels with the magnification.

Thank you. It’s corrected now.

  1. Throughout, the abbreviation DXT is unclear.

Thank you. It’s corrected now.

Reviewer 2 Report

The manuscript by Slyusarenko et al describe and evaluate a method for extravesicular nanovesicles (ENVs) from human plasma. The method utilizes the phenomenon of liquid-liquid phase portioning to achieve differential enrichment of ENVs from other blood plasma components. The simplicity of liquid-liquid portioning procedures renders this method particularly attractive. The procedure described in the manuscript would be of use to researchers in diverse fields exploring the role of EVs. However, several details are not clear and perhaps some interpretation of data optimistic. Some experimental controls are absent. The reviewer feels that the manuscript can be improved considerably to serve the readers better.

Specific comments:

  1. Fig1 – Materials and methods does not state the details of the DLS measurements. What is the reason for the extreme variation in size distributions observed in the three samples (3.5% PEG/1.5% DXT, 3.0% PEG/3.0% DXT, 2% PEG/2% DXT). It is unclear how the different ratio of the two components results in such a dramatic difference in the particles extracted from plasma. As the authors mention, EVs with large diameter > 1µM represent only a minor fraction of the population. What determines the selectivity for this specific enrichment of larger vesicles. What is the relative viscosity of the different LP1 fractions being compared here, and how was this measured and controlled for? Is it possible that the different composition (different viscosity) of the LP1 fractions contribute to the observed differences? There is no other corroborating evidence such as AFM or EM to demonstrate the presence of ENVs here.

  1. Fig 2: The colour scheme can be improved, to provide more contrast for the different components. It’s just not clear and does not make the point of selective enrichment of ENVs.

  1. The manuscript would have benefitted tremendously with specific quantitation of the markers which are both enriched in the LP phases and lost in the UP phases. How does this track with the multiple phase-partition cycles? What if a third or fourth portioning was performed? How many fold enrichment was achieved at each of these steps? While Fig 8 shows a dot blot of some markers, this is not quantitative. A table quantifying the enrichment/losses in the UP and LP would have been most useful. Discussion regarding data in fig 4 mentions [line 258] no loss of ENVs between the 1st and second portioning cycles (LP1 vs LP2), but the western dot blot data clearly shows loss of HAS, CD63, CD9 and HSP70 components (though the authors suggest this as a minor loss – paragraph with line 313].

  1. Fig 5; some “peaks” in the Raman spectra are highlighted, but there is no discussion as to what functional groups/vibrational modes these are attributed to. Without this it is unclear to the reader what is being tracked. For example, the peak at 1000 cm-1 not present in PEG or DEX. Were UC purified ENVs measured in PEG/DEX mixtures? The data here are not well described nor discussed. Using Raman spectra can the authors give an indication of the percent composition of the PEG/DEX components in the different UP and LP samples? The light microscopy data is not of much use in absence other control images – what is DEX-like, or PEG-like?

  1. AFM – This study would have been much stronger if controls, like the use of ultracentrifugation-depleted plasma samples were used in comparison with the actual samples. Both the AFM and Raman spectroscopy results would have been more convincing that what is being measured /visualized are bonafide ENVs. This is especially since we are not visualizing membrane components here, and the sample surface also shows the presence of smaller particles (attributed by the authors to dextran).

General comments: The manuscript would benefit considerably with a quantitation (of ENV markers and tracking RNAs), both to demonstrate more clearly what is going on with multiple extraction cycles, as well as for a direct comparison with the standard method of ultracentrifugation-based method. A table reporting fold purification of these components would have been useful. There are numerous typographical errors throughout the manuscript; Lines 105, 266, 311….and labeling in Fig3 Better controls are necessary to demonstrate that the particles in AFM are ENVs.

Author Response

Authors are very grateful to the Reviewer for a careful analysis of our manuscript and important comments.

The manuscript by Slyusarenko et al describe and evaluate a method for extravesicular nanovesicles (ENVs) from human plasma. The method utilizes the phenomenon of liquid-liquid phase portioning to achieve differential enrichment of ENVs from other blood plasma components. The simplicity of liquid-liquid portioning procedures renders this method particularly attractive. The procedure described in the manuscript would be of use to researchers in diverse fields exploring the role of EVs. However, several details are not clear and perhaps some interpretation of data optimistic. Some experimental controls are absent. The reviewer feels that the manuscript can be improved considerably to serve the readers better.

Specific comments:

  1. Fig1 – Materials and methods does not state the details of the DLS measurements. What is the reason for the extreme variation in size distributions observed in the three samples (3.5% PEG/1.5% DXT, 3.0% PEG/3.0% DXT, 2% PEG/2% DXT). It is unclear how the different ratio of the two components results in such a dramatic difference in the particles extracted from plasma. As the authors mention, EVs with large diameter > 1µM represent only a minor fraction of the population. What determines the selectivity for this specific enrichment of larger vesicles. What is the relative viscosity of the different LP1 fractions being compared here, and how was this measured and controlled for? Is it possible that the different composition (different viscosity) of the LP1 fractions contribute to the observed differences? There is no other corroborating evidence such as AFM or EM to demonstrate the presence of ENVs here.

We have extended description of DLS measurement and include information (methods and results) regarding measurement of suspensions viscosity. This parameter did not vary significantly over the set of measurements, so we can conclude that it did not interfere considerably with obtained results of DLS.

The main goal of this series of experiments was to select the optimal composition of the polymers. Therefore, we did not go deep into the mechanisms of particle size separation. Since the ratio of polymer concentrations was the only variable parameter in this experiment, it can be concluded that it is a factor responsible for the observed phenomenon. However, it should be borne in mind that size distributions are expressed in terms of the light intensity distribution, which depends on the size ~ R6. Thus, even a small number of large particles can cause a significant shift in the particle size distribution.

We definitely plan to investigate this observation in details and to analyze composition of LPs formed by different PTPS. One “work hypothesis” is formation of the multivesicular aggregates. We plat to use NTA and scanning electron microscopy to assay size, concentration and structure of large particles however it will be already next publication. 

  1. Fig 2: The colour scheme can be improved, to provide more contrast for the different components. It’s just not clear and does not make the point of selective enrichment of ENVs.

The scheme is intended to demonstrate a work-flow of procedure rather than principles of ENVs selective enrichment. Nonetheless, we have changed it by making symbols bigger.

  1. The manuscript would have benefitted tremendously with specific quantitation of the markers which are both enriched in the LP phases and lost in the UP phases. How does this track with the multiple phase-partition cycles? What if a third or fourth portioning was performed? How many fold enrichment was achieved at each of these steps? While Fig 8 shows a dot blot of some markers, this is not quantitative. A table quantifying the enrichment/losses in the UP and LP would have been most useful. Discussion regarding data in fig 4 mentions [line 258] no loss of ENVs between the 1stand second portioning cycles (LP1 vs LP2), but the western dot blot data clearly shows loss of HAS, CD63, CD9 and HSP70 components (though the authors suggest this as a minor loss – paragraph with line 313].

Thank you for this question. Indeed, quantification of markers distribution between two phases and during several phases partition is extremely interesting issue. However, exact tracking and quantification of certain markers are hampered by significant difference in volume and total protein concentration between the phases (“LP vs. UP” as well as in a range “LP1 vs. LP2 vs. LP3” and “UP1 vs. UP2 vs. UP3”). To demonstrate this difference, we included additional figure (n.7) that shows total protein concentration measured by Bradford assay. Considering difference in volume (LP was about 50-100 mkl while UP was about 1400-1450 mkl), total quantity of protein in upper phases was always much  higher while amount of proteins in LP was reducing with each round of separation. Due to concentration difference, we cannot use conventional western blot. Using dot blot assay we can roughly evaluate the concentration of major plasma protein – albumin (Figure 7b). Exosomal markers were not detectable in upper phases, therefore we checked them only in lower phases by dot blot.

So, we extended text and include new images to address this issue. However, as in Q1, our study was focused on evaluation fact ENV concentration in lower phase of PTPS rather than deep analysis of two-polymers system behavior, so we leave some issues for future investigations.

Thank you for correction of discussion section. Corresponding changes are made.

Fig 5; some “peaks” in the Raman spectra are highlighted, but there is no discussion as to what functional groups/vibrational modes these are attributed to. Without this it is unclear to the reader what is being tracked. For example, the peak at 1000 cm-1 not present in PEG or DEX. Were UC purified ENVs measured in PEG/DEX mixtures? The data here are not well described nor discussed. Using Raman spectra can the authors give an indication of the percent composition of the PEG/DEX components in the different UP and LP samples? The light microscopy data is not of much use in absence other control images – what is DEX-like, or PEG-like?

We are especially grateful for this question. As far as we know, attempt to apply RS for ENV analysis was done just once (Krafft et all., 2016, https://doi.org/10.1016/j.nano.2016.11.016), and protocol of such method is not yet established. Therefore, the set up of experiment and the data interpretation are still require further development and improvement. We introduced some changes in the text while most important issues are addressed below.

  • Functional group / peak assignment:

The first reason to use raman spectrometry was to explore similarity between UC-isolated ENV and PTPS-isolated ENV. This issue was answered since most characteristic UC-ENV peaks were observed in LP1 and PL2. In our opinion, assessment of the specific peaks in such a chemically complex composition as ENV (either UC- or PTPS-isolated ENV) would have approximate value. However, we now indicated possible inducer of ENV-specific peak at 1004 cm-1 (phenylalanine) and added a Supplementary Table 1 with possible assignment of other peaks in pure components spectra.

  • UC purified ENVs measured in PEG/DEX mixtures

No. We thank you for this idea of additional control, we will include it in our protocol. According to the previously established protocol, we presumed that measuring separately pure components (polymers PEG and dextran and ENVs isolated by UC) and products (UP1, LP1, UP2, LP2) would be enough. We estimated the minimum detection concentration for polymer presence by their dilution in water. But since now we will also make a comparison with a superposition (mixture) of pure components.

  • Indication of percent composition:

in principle, Raman spectroscopy is reported as a semi-quantitative and even quantitative method (for example, Rohleder et al, 2003 https://doi.org/10.1039/B408927H; Tan et al, 2017 https://doi.org/10.1021/acs.jafc.7b01814). But the current experiment is not able to provide such estimates for several reasons. It is worth noting that Raman spectroscopic setup is a one-ray system, therefore in each time moment a spectrum of just one sampling dot is recorded. Meanwhile, as it was demonstrated on UP2 sample, different components may have strong phase separation - this is the first limitation. The second limitation is that the analyzed assay is a dried drop, that has inhomogeneous distribution of analyte on its area, that was especially noticeable on UP2 sample, as demonstrated on Figure 6. Map (screening) measurement of the whole sample could be in principle provided on the Raman equipment used, but it would demand a great multitude of sampling points. As a drop should have a size such that a sufficient concentration of analyte is found in the laser spot, the area of the dried drop (diameter is around 2.5 mm) is much greater than the objective field of view which is around 100x120 um in 50x objective, which is used because a 10x objective can’t provide a sufficient power density. Therefore, technically this multitude of sampling points has to be divided into a set of series, covering all the drop by a series of spectra in each field of view. The third limitation is that such experiment requires proper calibration during the whole measurement series.

  • Light microscopy data

The raman spectra of LPs were well reproducible while UPs assessment gave as two different types of spectra: one was similar to spectra of pure PEG, another was similar to spectra of pure DEX. Light microscopy images were included in the manuscript as a possible explanation of two types raman spectra repeatedly measured in upper phases. Microscopy allowed as to associate the two different spectra with different areas of dried drops. These images allowed to observe separation of polymers within the suspension that is not visible in tube. Now we change the Figure 6: indicated area of magnitude and included in the Figure images of dried drops of pure polymers solutions.

In addition, we change a color of the Figure 5 to make it compatible with Figure 4 and Figure 7.

  1. AFM – This study would have been much stronger if controls, like the use of ultracentrifugation-depleted plasma samples were used in comparison with the actual samples. Both the AFM and Raman spectroscopy results would have been more convincing that what is being measured /visualized are bonafide ENVs. This is especially since we are not visualizing membrane components here, and the sample surface also shows the presence of smaller particles (attributed by the authors to dextran).

We agree with this statement. Of course, visualization of the vesicular structure and plasma membrane would be the most convincing conformation of the vesicular nature of the particles isolated by PTPS. We are still working on establishing a protocol for Cryo-EM and it is still difficult due to the presence of dextran. Of course, this is a reasonable request to evaluate YaK depleted plasma as a negative control. We did not include these samples in AFM analysis only because we used the standard, well-established UC protocol to isolate ENV from plasma. We previously confirmed the quality of isolated vesicles (using Cryo-EM and AFM) (1-3), and in this study we did not take care of negative controls. Instead, we used UC-ENV as a kind of positive control for vesicles isolated by PTPS. Unfortunately, we will not be able to repeat AFM in the next future and analyze a plasma depleted in UC. However, we will keep this suggestion in mind and will have this control for the next experiments.

General comments: The manuscript would benefit considerably with a quantitation (of ENV markers and tracking RNAs), both to demonstrate more clearly what is going on with multiple extraction cycles, as well as for a direct comparison with the standard method of ultracentrifugation-based method. A table reporting fold purification of these components would have been useful.

As it mentioned above, ENV markers are too diluted in upper phases, so we were not able to track them using conventional approaches (western blot or dot blot). As for RNA tracking, we used several commercial kit commonly used for ENV isolation and further RNA/miRNA analysis. Since analysis of non-vesicular RNA/miRNA was beyond of our scope, we did not include this experiment in our study.

There are numerous typographical errors throughout the manuscript; Lines 105, 266, 311….and labeling in Fig3

Corrected.

We thank again to Reviewer for careful evaluation of our manuscript!

  1. Shtam T, Evtushenko V, Samsonov R, Zabrodskaya Y, Kamyshinsky R, Zabegina L, et al. Evaluation of immune and chemical precipitation methods for plasma exosome isolation. PLoS One. 2020;
  2. Shtam T, Naryzhny S, Samsonov R, Karasik D, Mizgirev I, Kopylov A, et al. Plasma exosomes stimulate breast cancer metastasis through surface interactions and activation of FAK signaling. Breast Cancer Res Treat [Internet]. 2019 Feb 27;174(1):129–41. Available from: http://link.springer.com/10.1007/s10549-018-5043-0
  3. Shtama Т, Naryzhny S, Kopylovd A, Petrenko E, Samsonov R, Kamyshinsky R, et al. Functional Properties of Circulating Exosomes Mediated by Surface-Attached Plasma Proteins. J Hematol. 2018;7(4):149–53.

Round 2

Reviewer 2 Report

The manuscript does have some improvement in presentation, and it is understandable that the authors are unable to provide additional experimentation to support their results. Addition of these controls would have definitely solidified the conclusions and improved the quality of the manuscript considerably. As it stands, the manuscript suggests readers an interesting avenue to explore in their own EV research, though it does not provide a thoroughly characterized and detailed protocol. In light of the short comings, addition of some discussion by the authors suggesting which steps a reader might optimize, might be of use.